# Grad2Task: Improved Few-shot Text Classification Using Gradients for Task Representation

**Jixuan Wang**[1,2,3]  **Kuan-Chieh Wang**[1,2]  **Frank Rudzicz**[1,2,4]  **Michael Brudno**[1,2,3]

[1]University of Toronto, [2]Vector Institute, [3]University Health Network, [4]Unity Health Toronto
{jixuan, wangkua1, frank, brudno}@cs.toronto.edu

## Abstract

Large pretrained language models (LMs) like BERT have improved performance in many disparate natural language processing (NLP) tasks. However, fine tuning such models requires a large number of training examples for each target task. Simultaneously, many realistic NLP problems are "few shot", without a sufficiently large training set. In this work, we propose a novel conditional neural process-based approach for few-shot text classification that learns to transfer from other diverse tasks with rich annotation. Our key idea is to represent each task using gradient information from a base model and to train an adaptation network that modulates a text classifier conditioned on the task representation. While previous task-aware few-shot learners represent tasks by input encoding, our novel task representation is more powerful, as the gradient captures input-output relationships of a task. Experimental results show that our approach outperforms traditional fine-tuning, sequential transfer learning, and state-of-the-art meta learning approaches on a collection of diverse few-shot tasks. We further conducted analysis and ablations to justify our design choices.

## 1 Introduction

Transformer-based pretrained large-scale language models (LMs) have achieved tremendous success on many NLP tasks [15, 34], but require a large number of in-domain labeled examples for fine-tuning [54]. One approach to alleviate that issue is to fine-tune the model on an intermediate (source) task that is related to the target task. While previous work has focused on this setting for a single source task [33, 50], the problem of transferring from a set of source tasks has not been thoroughly considered. Wang et al. [51] showed that a naïve combination of multiple source tasks may negatively impact target task performance. Additionally, transfer learning methods typically consider the setting where more than a medium (N>100) number of training examples are available for both the source and target tasks.

In this work we develop a method to improve a large pre-trained LM for few-shot text classification problems by transferring from multiple source tasks. While it is possible to directly adapt recent advances in meta-learning developed for few-shot image classification, a unique challenge in the NLP setting is that source tasks do no share the same task structure. Namely, on standard few-shot image classification benchmarks, the training tasks are sampled from a "single" larger dataset, and the label space contains the same task structure for all tasks. In contrast, in text classification tasks, the set of source tasks available during training can range from sentiment analysis to grammatical acceptability judgment. Different source tasks could not only be different in terms of input domain, but also their task structure (i.e. label semantics, and number of output labels).

This challenging problem requires resistance to overfitting due to its few-shot nature and more task-specific adaptation due to the distinct nature among tasks. Fine-tuning based approaches are known to suffer from the overfitting issue and approaches like MAML [17] and prototypical networks

35th Conference on Neural Information Processing Systems (NeurIPS 2021).

(ProtoNet) [41] were not designed to meta-learn from such a diverse collection of tasks. CNAP [37], a conditional neural process [18], explicitly designed for learning from heterogeneous task distributions by adding a task conditional adaptation mechanism is better suited for solving tasks with a shifted task distribution. We design our approach based on the CNAP framework. Compared with CNAP, we use pretrained transformers for text sequence classification instead of convolutional networks for image classification. We insert adapter modules [21] into a pretrained LM to ensure efficiency and maximal parameter-sharing, and learn to modulate those adapter parameters conditioned on tasks, while CNAP learns to adapt the whole feature extractor. Additionally, we use gradient information for task representation as opposed to average input encoding in CNAP, since gradients can capture information from both input and ouput space.

In summary, our main contributions and findings are:

- We propose the use of gradients as features to represent tasks under a model-based meta-learning framework. The gradient features are used to modulate a base learner for rapid generalization on diverse few-shot text classification tasks.

- We use pre-trained BERT with adapter modules [21] as the feature extractor and show that it works better than using only the BERT model on few-shot tasks, while also being more efficient to train as it has much fewer parameters to learn.

- We compare our approach with traditional fine-tuning, sequential transfer learning, and state-of-the-art meta-learning approaches on a collection of diverse few-shot text classification tasks used in [5] and three newly added tasks, and show that our approach achieves the best performance. Our codes are publicly available[1].

## 2 Related Work

### 2.1 Few-shot text classification

There is a wide range of text classification tasks, such as user intent classification, sentiment analysis, among others. Few-shot text classification tasks refer to those that contain novel classes unseen in training tasks and where only a few labeled examples are given for each class [55]. Meta-learning approaches have been proposed for this problem [7, 19, 5]. To increase the amount and diversity of training episodes, previous work trained few-shot learners by semi-supervised learning [40] and self-supervised learning [6]. We compare different meta-learning approaches following the experiment procedures of [5].

### 2.2 Meta-learning

Metric-based meta-learning approaches [49, 41, 45] try to learn embedding models on which simple classifiers can achieve good performance. Optimization-based approaches [17, 38, 30] are designed to learn good parameter initialization that can quickly adapt to new tasks within a few gradient descent steps. "Black box" or model-based meta-learning approaches use neural networks to embed task information and predict test examples conditioned on the task information [39, 36, 28, 31, 18, 37]. Our approach falls into this third category and is mostly related to CNAP [37]. Compared with CNAP we use a different feature extractor and different mechanisms for modulation, as well as focusing on different problems. An important difference with CNAP is that we use gradient information as features to represent tasks, instead of using average input encoding. Gradient information has also been used to generate task-dependent attenuation of parameter initialization under the MAML framework [4].

### 2.3 Transfer learning

Our work is also related to transfer learning. Instead of fine-tuning a pretrained language model directly on a target task, Phang et al. [33] and Talmor and Berant [46] showed that intermediate training on data-rich supervised tasks is beneficial for downstream performance. Wang et al. [51] found that there is no guarantee that multi-task learning or intermediate training yields better performance than direct fine-tuning on target tasks. Transferability estimation is the task of predicting whether it is

---

[1] https://github.com/jixuan-wang/Grad2Task

beneficial to transfer from some task to another one. There have been works estimating transferability of NLP tasks based on handcrafted features [8, 23], while data-driven approaches have been studied mainly for computer vision tasks [56, 47, 43, 29].

TASK2VEC is a task embedding approach based on the Fisher Information Matrix (FIM) [2]. Transferability between tasks can be easily evaluated by the distance between their task embeddings according to some simple metric. In the NLP domain, given a target task and a predefined set of source tasks, Vu et al. [50] proposed to use task embedding based on FIM to measure task similarity and predict transferability according to the distance between task embeddings. We draw inspiration from this work to use gradients as features for task representation. Instead of one-to-one transfer, our work can be seen as transferring from a set of source tasks to new tasks through meta-learning.

## 3 Problem Definition

Following the terminologies in meta learning, each task (or episode), $t = (\mathcal{S}, \mathcal{Q})$, is specified by a support set (training set) $\mathcal{S}$ and a query set (test set) $\mathcal{Q}$, where $\mathcal{S} = \{(x_i, y_i)\}_{i=1}^{|\mathcal{S}|}$, $\mathcal{Q} = \{(x_i, y_i)\}_{i=1}^{|\mathcal{Q}|}$, $x_i$ is a text sequence and $y_i$ is a discrete value corresponding to a class. Different tasks could have different input domains, different numbers of classes, different label space, and so on.

Our goal is to train a text classification model on a set of labeled datasets $\{\mathcal{D}_i^s\}_{i=1}^N$ of $N$ source tasks. The model is expected to achieve good performance on new target tasks after training. For each target task, a small labeled dataset, e.g., five shots for each class, is given for adaptation or fine-tuning and the full test dataset is used for evaluation. Note that when learning on target tasks, we assume no access to the source tasks nor other target tasks. This is different with multi-task learning where target tasks are learned together with source tasks.

## 4 Model Design

To handle diverse tasks with different structures a single model may not be flexible enough. Thus, for every task, we utilize a task embedding network to capture the task nature and adapt a base model conditioned on the current task. Task-specific adaptation is done by generating shifting and scaling parameters, named adaptation parameters, that are applied on the hidden representations inside the base model. Figure 1 shows an overview of our model architecture, mainly consisting of three parts: a base model denoted by $f$, a task embedding network denoted by $d$, and an adaptation network denoted by $a$.

To make a prediction, our model takes the following steps: 1) given a support set, it uses the base model (Section 4.1) to compute gradients w.r.t. a subset of its parameters to use as input to the task embedding network (Section 4.2), 2) the task embedding network maps these gradients to task representations, 3) layer-wise adaptation networks (Section 4.3) take as input the task representations and output adaptation parameters, and lastly 4) the adaptation parameters are applied to the base model before it predicts on the query set.

The training of our model consists of two stages: 1) we first train the base model episodically as a prototypical network; 2) we then freeze the base model and train the task embedding network and the adaptation network using the same loss function of the first stage (Section 4.4).

Next we introduce our model in details. The list of notations used throughout the paper can be found in Appendix A.

### 4.1 Base Model: BERT & Bottleneck Adapters

Our base model is built upon transformer-based pretrained LMs. We use the pretrained BERT$_{\text{BASE}}$ model throughout this paper, although other transformer-based pretrained LMs are also applicable [35, 25]. Following Houlsby et al. [21], we insert two adapter modules containing bottleneck layers into each transformer layer of BERT. Each bottleneck adapter is denoted as $\alpha_l, l = 1...2L$ where L is the total number of transformer layers in BERT. We use the bottleneck adapters because, first, BERT with adapters are more efficient to train and less vulnerable to overfitting – a desirable property in the few-shot setting. Additionally, this simplifies our goal of using gradients for task representation. It is infeasible to use the gradients of the whole BERT model because the dimensionality is too large.

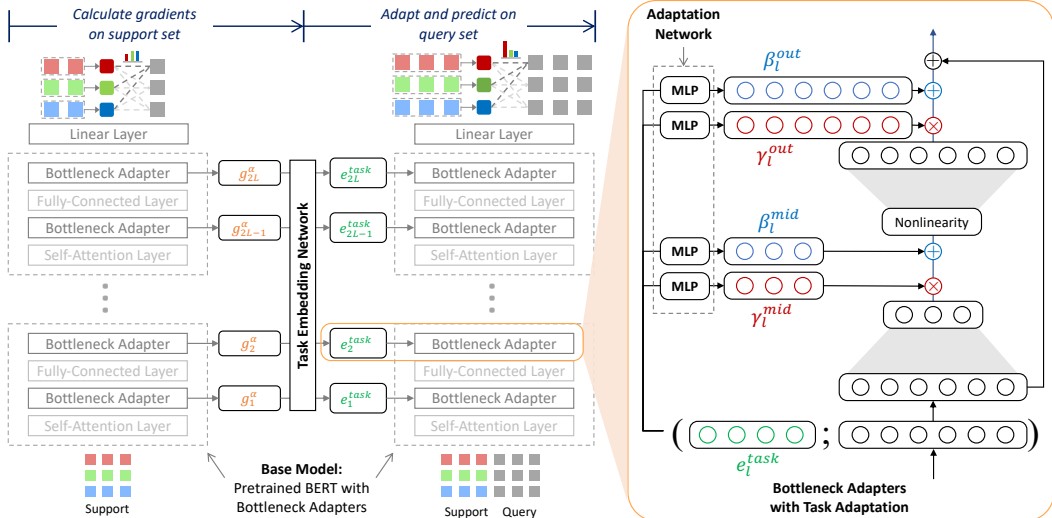

Figure 1: Model architecture overview. Left: The **base model** contains a pretrained BERT model with bottleneck adapters inserted in each transformer layer and a linear layer stacked on the top. The **task embedding network** is an RNN-based model that maps gradients from the base model into task representations in a layer-wise fashion, which are used for base model adaptation. Right: The **adaptation network** contains MLPs that take as input the task representations and intermediate activate inside the base model, and outputs shifting and scaling parameters that are used to adapt the base model. Specifically, given the task embedding for a certain layer, we concatenate it with the input hidden representation to the current layer and map them into four shifting/scaling parameters by four MLP networks, which are applied on the hidden and output representations of the adapter, referred as the auto-regressive adaptation.

Instead, we compute gradients w.r.t. parameters in these bottleneck layers ($< 0.5\%$ of the number of parameters in BERT).

Since text classification is a sequence classification task, we need a pooling method to represent each sequence by a single embedding before the embedding is fed into a classifier. Following Devlin et al. [15], this single embedding is obtained by applying a linear transformation on top of the contextual embedding of the special '[CLS]' token inserted at the beginning of every sequence. We further apply a fully connected layer on top of the BERT output. Together, we name the pretrained BERT with adapters and the last linear layer as the 'base model', denoted by $f$. We denote the parameters of the pretrained BERT as $\theta$, bottleneck adapters as $\{\alpha_l\}_{l=1}^{2L}$, and linear layer as $\omega$.

For the classifier, we use the ProtoNet classifier [41]. Given a support set $\mathcal{S}$, a query example $x$ and our base model $f$, a nearest cluster classifier can be described as follows:

$$p^{\text{base}}(y = c \,|\, x) = \text{softmax}(\text{euc}(f(x; \theta, \alpha, \omega), \mu^c)), \tag{1}$$

$$\mu^c = \frac{1}{|\mathcal{S}^c|} \sum_{x_i \in \mathcal{S}^c} f(x_i; \theta, \alpha, \omega)$$

where $\mathcal{S}^c = \{(x_i, y_i) \,|\, (x_i, y_i) \in \mathcal{S}, y_i = c\}$, $\mu^c$ is the cluster center in the embedding space for support examples with class $c$, and euc refers to the Euclidean distance.

## 4.2 Task Embedding Network for Per-Layer Task Encoding

Our task embedding network, denoted by $d$, is parametrized as a recurrent neural network (RNN) over the layers. We use an RNN to summarize the gradient information from the lower layers so that the higher layers can better adapt to the current tasks. Also, using RNNs can enable parameter sharing across different layers. Concatenating gradients of all layers as input will result in an extremely large task embedding model. The task embedding network takes as input at each layer the gradient information of the adapter parameters at the given layer. Following Achille et al. [2], we use the

gradient information defined by the FIM of the base model's parameters, denoted by $\Theta = \{\theta, \alpha, \omega\}$. The FIM is computed as:

$$F_\Theta = \mathbb{E}_{x,y \sim \hat{p}(x)p^{\text{base}}(y \mid x)}[\nabla_\Theta \log p^{\text{base}}(y \mid x) \nabla_\Theta \log p^{\text{base}}(y \mid x)^T] \tag{2}$$

where $\hat{p}(x)$ is the empirical distribution defined by the dataset. Following Achille et al. [2], we only use the diagonal values of $F_\Theta$. We only use those values corresponding to the adapter parameters, denoted by $g_l^\alpha$ for adapter layer $l$, $l = 1 \ldots 2L$.

Then, the RNN-based task embedding network $d$, parameterized by $\phi$, maps each $g_l^\alpha$ into a task embedding:

$$e_l^{task} = d^o(h_l; \phi), \tag{3}$$
$$h_l = d^h(g_l^\alpha, h_{l-1}; \phi)$$

where $h_l$ is the hidden representation at layer $l$, $e_l^{task}$ is the task embedding at layer $l$, $h_0$ is the learnable initial RNN hidden state, and $d^o$ and $d^h$ are the functions of $d$ that produce output and hidden states, respectively.

### 4.3 Adaptation Network with Auto-Regressive Adaptation

As shown on the right side of Figure 1, at each layer $l$, the adaptation network output four adaptation parameters: $\gamma_l^{mid}$ and $\beta_l^{mid}$ are scaling and shifting adaptation parameters applied on the hidden representation after the middle layer, and $\gamma_l^{out}$ and $\beta_l^{out}$ are scaling and shifting adaptation parameters applied on the output hidden representation. The input to the adaptation network at each layer consists of both the task representation and the intermediate activation output by previous adapted layers. Due to the conditioning on previous modulated layers, we refer this adaptation method as auto-regressive adaptation, following Requeima et al. [37].

Starting from the input layer, the adaptation network $a$ produces adaptation parameters for the first adapter layer as follows:

$$\gamma_1^{mid}, \beta_1^{mid}, \gamma_1^{out}, \beta_1^{out} = a(e_1^{task}, x; \psi) \tag{4}$$

where $x$ is input to the base model. We denote the bottleneck adapter network after modulation as $\alpha'$.

For the subsequent layers, the adaptation network takes as input the combination of the task embedding and intermediate activation of the adapted base model, and outputs adaptation parameters as follows:

$$\gamma_l^{mid}, \beta_l^{mid}, \gamma_l^{out}, \beta_l^{out} = a(e_l^{task}, f_{l-1}(x; \theta, \{\alpha_j'\}_{j=1}^{l-1}); \psi), \quad l = 2 \ldots 2L \tag{5}$$

where $f_{l-1}(x; \theta, \{\alpha_j'\}_{j=1}^{l-1})$ denotes the intermediate activation before layer $l$ from the base model with the first $l - 1$ adapters that have been modulated. Finally, the prediction from our modulated base model is written as:

$$p^{\text{final}}(y = c \mid x) = \text{softmax}(\text{euc}(f(x; \theta, \alpha', \omega), \mu^c)), \tag{6}$$
$$\mu^c = \frac{1}{|S^c|} \sum_{x_i \in \mathcal{S}^c} f(x_i; \theta, \alpha', \omega)$$

Notice that the only difference between $p^{\text{final}}(y = c \mid x)$ and $p^{\text{base}}(y = c \mid x)$ is using $\alpha'$ or $\alpha$, i.e. our task-specific modulation.

Our adaptation mechanism is similar with the FiLM layer [32] but different in terms of that here we only apply adaptation on the hidden representation of the "[CLS]", since only its embedding is used as the sequence embedding. This is analogous to only adapting the the first channel of each hidden representation to a Film layer. We also tried adapting embeddings of all tokens but performance is worse, as shown in Section 6.2.

### 4.4 Model Training

Similar with CNAP [37], we train our model in two stages. In the first stage, we train the base model using episodic training [41]. The training algorithm is showed in Appendix B. Given an episode $t = (\mathcal{S}, \mathcal{Q})$, the loss is computed as:

$$\ell_{pn}(t) = \frac{1}{|\mathcal{Q}|} \sum_{x_i \in \mathcal{Q}} -\log p^{\text{base}}(y_i \mid x_i) \tag{7}$$

**Algorithm 1:** Training the task embedding network and adaptation network for quick adaptation to new tasks.

---

**Input** : $T$: A set of text classification tasks for training. $\mathcal{D}_i$: The labeled dataset of each task $i$. The base model trained in the first stage.

**Output :** $\phi$: Parameters of the task embedding model. $\psi$: Parameters of the adaptation network.

---

**1** **while** *not converge* **do**

**2** $\quad$ Sample $M$ episodes $\{t_i\}_{i=1}^M$, each episode $t_i = (\mathcal{S}_i, \mathcal{Q}_i), \mathcal{S}_i \subseteq \mathcal{D}_i, \mathcal{Q}_i \subseteq \mathcal{D}_i$

**3** $\quad$ **for** $i \leftarrow 1$ **to** $M$ **do**

**4** $\quad\quad$ **for** $s \leftarrow 1$ **to** $S$ **do**

**5** $\quad\quad\quad$ Sample a few examples for each class from $\mathcal{S}_i$ and build prototypes $\{\mu^c\}$

**6** $\quad\quad\quad$ Sample another subset of examples $\hat{\mathcal{S}}_i \subseteq \mathcal{S}_i$

**7** $\quad\quad\quad$ For each $(x_j, y_j) \in \hat{\mathcal{S}}_i$, sample $y_j' \sim p^{\text{base}}(y \,|\, x_j)$ `// `$p^{\text{base}}$` is given by Equation (`1`)`

**8** $\quad\quad\quad$ $g_s^\alpha = -\sum_{j=1}^{|\hat{\mathcal{S}}_i|} \left( \nabla_\alpha \log p^{\text{base}}(y = y_j' \,|\, x_j) \right)^2, (x_j, y_j) \in \hat{\mathcal{S}}_i$ `  // calculating gradient`
$\quad\quad\quad\quad\quad\quad\quad\quad\quad\quad\quad\quad\quad\quad\quad\quad\quad\quad\quad\quad\quad\quad\quad$ `// information as in Equation (`2`)`

**9** $\quad\quad$ **end**

**10** $\quad\quad$ $g^\alpha = \frac{1}{S} \sum_{s=1}^S g_s^\alpha$

**11** $\quad\quad$ $e^{task} = d(g^\alpha; \phi)$ $\quad\quad\quad\quad\quad\quad\quad\quad\quad\quad\quad$ `// generating task representation`

**12** $\quad\quad$ $\gamma^{in}, \beta^{in}, \gamma^{out}, \beta^{out} = a(e^{task}; \psi)$ $\quad\quad\quad$ `// generating adaptation parameters`

**13** $\quad\quad$ $\ell_i = \frac{1}{|\mathcal{Q}_i|} \sum_{j=1}^{|\mathcal{Q}_i|} -\log p^{\text{final}}(y = y_j | x_j)$ $\quad\quad$ `// calculating the loss on query set`
$\quad\quad\quad\quad\quad\quad\quad\quad\quad\quad\quad\quad\quad\quad\quad\quad\quad\quad\quad\quad\quad\quad\quad$ `// `$p^{\text{final}}$` is given by Equation (`6`)`

**14** $\quad$ **end**

**15** $\quad$ $\{\phi, \psi\} \leftarrow \{\phi, \psi\} - lr \cdot \nabla_{\{\phi,\psi\}} \sum_{i=1}^M \ell_i$ $\quad\quad\quad\quad$ `// gradient descent on `$\phi$` and `$\psi$

**16** **end**

**17** **return** $\phi, \psi$

---

During this stage, only the parameters of the bottleneck adapters, layer normalization parameters and the top linear layer are updated while other parameters of the base model are frozen. The parameters are learned by:

$$\alpha^*, \omega^* = \underset{\alpha, \omega}{\arg\min} \mathbb{E}_{t \in T}[\ell_{pn}(t)] \tag{8}$$

In the second stage, the task embedding network and the adaptation network are trained episodically to generate good quality task embeddings and adaptation parameters for better performance on the query set of each episode. Specifically, as shown in Algorithm 1, we freeze the encoding network and only train the task embedding network $d$ and the adaptation network $a$. In this stage, the frozen base model is used to generate task-specific gradient information and also be adapted to predict the query labels. Same with the first training stage, we also use the ProtoNet loss to train $d$ and $a$ but use the adapted based model. The parameters of the task embedding network $\phi$ and the adaptation network $\psi$ is learned by:

$$\phi^*, \psi^* = \underset{\phi, \psi}{\arg\min} \mathbb{E}_{t \in T}[\ell_{pn}'(t)], \tag{9}$$

$$\ell_{pn}'(t) = \frac{1}{|\mathcal{Q}|} \sum_{x_i \in \mathcal{Q}} -\log p^{\text{final}}(y_i \,|\, x_i)$$

where the loss is computed based on the modulated model.

## 5 Experiments and Results

### 5.1 Experiment setup

We use datasets for training and testing that have different input domains and different numbers of labels. Each dataset appears during either training or testing – not both. Following [5], we use tasks from the GLUE benchmark [52] for training. Specifically, we use WNLI (m/mm), SST-2, QQP, RTE, MRPC, QNLI, and the SNLI dataset [10], to which we refer as our 'meta-training datasets'. The validation set of each dataset is used for hyperparameter searching and model selection. We train

our model and other meta-learning models by sampling episodes from the meta-training tasks. The sampling process first selects a dataset and then randomly selects $k$-shot examples for each class as the support set and another $k$-shot as the query set. As with Bansal et al. [5], the probability of a task being selected is proportional to the square root of its dataset size.

We use the same test datasets as Bansal et al. [5], but also add several new datasets to increase task diversity, namely the Yelp [1] dataset to predict review stars, the SNIPS dataset [13] for intent detection, and the HuffPost [26] dataset for news headline classification. We refer to this set of tasks as our 'meta-testing datasets'. For each test task and a specific number of shot $k$, ten $k$-shot datasets are randomly sampled. Each time one of the ten datasets is given for model training or fine-tuning, the model is then tested on the full test dataset of the corresponding task. This is in line with real world applications where models built on a small training set are tested on the full test set.

Several test datasets used by Bansal et al. [5] contain very long sentences, but they limited the maximal sequence length to 128 tokens in their experiments. Truncating long sequences beyond the maximal length might lose important information for text classification, leading to noise in the test results. In order to ensure the results can truly reflect text classification, we discard a few datasets used by Bansal et al. [5] that contain many very long sentences.

See Appendix C for more details about the datasets we use. See Appendix G.1 for the results on all datasets used by Bansal et al. [5].

## 5.2 Few shot text classification results

We use the cased version of the BERT$_{\text{BASE}}$ model for all experiments. We compare our proposed approach with the following approaches:

**BERT.** The pretrained BERT is simply fine-tuned on the labeled training data of the target task and then evaluated on the corresponding test data. No transfer learning happens with this model.

**MT-BERT.** The pretrained BERT is first trained on the meta-training tasks via multi-task learning and then fine-tuned and evaluated on each test task.

**ProtoNet-BERT.** ProtoNet using pretrained BERT plus a linear layer as the feature extractor. The model is trained episodically on the meta-training datasets.

**ProtoNet-BN.** Similar with ProtoNet-BERT but with adapter modules inserted in the transformer layers. Only the parameters of the adapters, linear layer, and layer normalization layers inside BERT are updated during training. The model is trained episodically on the meta-training datasets.

**MAML based approach.** We compare with the MAML-based approach, named Leopard, proposed by Bansal et al. [5] with first-order approximation and meta-learned per-layer learning rates. We re-implemented this model, and include both their reported results and results of our implementation for fair comparison. See Appendix D.1 for more details.

**Grad2Task.** This is our proposed model built upon pretrained ProtoNet-BN and trained episodically to learn to adapt.

Results are shown in Table 1. Overall, our proposed method achieves the best performance. Our model is built upon ProtoNet-BN but keeps its parameters untouched and only learns to adapt the bottleneck adapter modules for different tasks. On average, our approach improves over ProtoNet-BN by 1.02%, indicating task conditioning can further improve a strong baseline.

Surprisingly, we find that ProtoNet-BERT achieves very good performance and outperforms the optimization-based method, Leopard, while Bansal et al. [5] reported the opposite results. We thus re-implemented both Leopard and ProtoNet-BERT and confirm that our MAML implementation has similar performance with Leopard but our ProtoNet-BERT results are much better than theirs (see Appendix D.1 for a more detailed comparison).

Although we observe that ProtoNet-BERT has better performance and faster convergence rates during training and validation, it is outperformed by ProtoNet-BN which has orders of magnitude fewer parameters to learn (fewer than 0.5% of the number of BERT's parameters). We hypothesize this is because ProtoNet-BERT is more vulnerable to overfitting on the meta-training tasks.

See Appendix D.2 for the details of implementation, model size and training efficiency. We also compare with other fine-tuning approaches, like further fine-tuning a trained ProtoNet. See Appendix G.2 for the results.

Table 1: Results on diverse few-shot text classification tasks. Results marked with '*' are from [5]. For Leopard, we reuse the their results on the first seven tasks and report the results of our implementation on the last three newly added tasks.

| | Model | BERT* | | MT-BERT* | | Leopard | | PN-BERT | | PN-BN | | Grad2Task | |
|---|---|---|---|---|---|---|---|---|---|---|---|---|---|
| # | | Mean | Std | Mean | Std | Mean | Std | Mean | Std | Mean | Std | Mean | Std |
| | airline | 42.76 | 13.50 | 46.29 | 12.26 | 54.95 | 11.81 | 65.39 | 12.73 | 65.33 | 7.95 | **70.64** | 3.95 |
| | disaster | **55.73** | 10.29 | 50.61 | 8.33 | 51.45 | 4.25 | 54.01 | 2.90 | 53.48 | 4.76 | 55.43 | 5.89 |
| | emotion | 9.20 | 3.22 | 9.84 | 2.14 | 11.71 | 2.16 | 11.69 | 1.87 | 12.52 | 1.32 | **12.76** | 1.35 |
| | political_audience | 51.89 | 1.72 | 51.53 | 1.80 | 52.60 | 3.51 | **52.77** | 5.86 | 51.88 | 6.37 | 51.28 | 5.74 |
| | political_bias | 54.57 | 5.02 | 54.66 | 3.74 | 60.49 | 6.66 | 58.26 | 10.42 | **61.72** | 5.65 | 58.74 | 9.43 |
| 4 | political_message | 15.64 | 2.73 | 14.49 | 1.75 | 15.69 | 1.57 | 17.82 | 1.33 | 20.98 | 1.69 | **21.13** | 1.97 |
| | rating_kitchen | 34.76 | 11.20 | 36.77 | 10.62 | 50.21 | 9.63 | **58.47** | 11.12 | 55.99 | 9.85 | 57.09 | 9.74 |
| | huffpost_10 | - | - | - | - | 11.8 | 1.41 | 14.97 | 1.69 | 16.81 | 2.52 | **18.5** | 2 |
| | snips | - | - | - | - | 21.36 | 2.7 | 28.99 | 3.93 | 46.29 | 3.91 | **52.51** | 2.68 |
| | yelp | - | - | - | - | 36.95 | 2.98 | 42.84 | 2.66 | 42.64 | 2.93 | **43** | 3.55 |
| | **Average** | - | - | - | - | 36.72 | 4.67 | 40.52 | 5.45 | 42.76 | 4.70 | **44.11** | 4.63 |
| | airline | 38.00 | 17.06 | 49.81 | 10.86 | 61.44 | 3.90 | 69.14 | 4.84 | 69.37 | 2.46 | **72.04** | 2.58 |
| | disaster | 56.31 | 9.57 | 54.93 | 7.88 | 55.96 | 3.58 | 54.48 | 3.17 | 53.85 | 3.03 | **57.49** | 5.36 |
| | emotion | 8.21 | 2.12 | 11.21 | 2.11 | 12.90 | 1.63 | 13.10 | 2.64 | 13.87 | 1.82 | **13.99** | 1.90 |
| | political_audience | 52.80 | 2.72 | 54.34 | 2.88 | 54.31 | 3.95 | **55.17** | 4.28 | 53.08 | 6.08 | 52.60 | 5.55 |
| | political_bias | 56.15 | 3.75 | 54.79 | 4.19 | 61.74 | 6.73 | 63.22 | 1.96 | **65.36** | 2.03 | 64.06 | 1.12 |
| 8 | political_message | 13.38 | 1.74 | 15.24 | 2.81 | 18.02 | 2.32 | 20.40 | 1.12 | **21.64** | 1.72 | 21.31 | 1.16 |
| | rating_kitchen | 34.49 | 8.72 | 47.98 | 9.73 | 53.72 | 10.31 | 57.08 | 11.54 | 56.27 | 10.70 | **58.35** | 9.83 |
| | huffpost_10 | - | - | - | - | 12.73 | 2.23 | 16.52 | 1.48 | 19.03 | 2.18 | **21.12** | 1.69 |
| | snips | - | - | - | - | 20.51 | 2.93 | 32.19 | 1.85 | 52.74 | 2.74 | **57.19** | 2.77 |
| | yelp | - | - | - | - | 38.31 | 3.52 | **44.7** | 1.68 | 43.83 | 2.45 | 43.66 | 1.65 |
| | **Average** | - | - | - | - | 38.96 | 4.11 | 42.60 | 3.46 | 44.90 | 3.52 | **46.18** | 3.36 |
| | airline | 58.01 | 8.23 | 57.25 | 9.90 | 62.15 | 5.56 | 71.06 | 1.60 | 69.83 | 1.80 | **72.30** | 1.75 |
| | disaster | **64.52** | 8.93 | 60.70 | 6.05 | 61.32 | 2.83 | 55.30 | 2.68 | 57.38 | 5.25 | 59.63 | 3.11 |
| | emotion | 13.43 | 2.51 | 12.75 | 2.04 | 13.38 | 2.20 | 12.81 | 1.21 | 14.11 | 1.12 | **13.72** | 1.24 |
| | political_audience | **58.45** | 4.98 | 55.14 | 4.57 | 57.71 | 3.52 | 56.16 | 2.81 | 57.23 | 2.77 | 55.46 | 3.34 |
| | political_bias | 60.96 | 4.25 | 60.30 | 3.26 | 65.08 | 2.14 | 61.98 | 6.89 | **65.38** | 1.71 | 63.83 | 0.74 |
| 16 | political_message | 20.67 | 3.89 | 19.20 | 2.20 | 18.07 | 2.41 | 21.36 | 0.86 | **24.00** | 1.39 | 22.22 | 1.20 |
| | rating_kitchen | 47.94 | 8.28 | 53.79 | 9.47 | 57.00 | 8.69 | 61.00 | 9.17 | 59.45 | 8.33 | **61.72** | 6.38 |
| | huffpost_10 | - | - | - | - | 13.78 | 1.05 | 17.74 | 1.42 | 21.43 | 1.53 | **23.57** | 1.76 |
| | snips | - | - | - | - | 24.49 | 1.62 | 33.84 | 3.08 | 54.64 | 2.06 | **59.47** | 1.91 |
| | yelp | - | - | - | - | 39.7 | 2.31 | **45.5** | 1.72 | 44.78 | 1.8 | 44.87 | 2.09 |
| | **Average** | - | - | - | - | 41.27 | 3.23 | 43.67 | 3.14 | 46.82 | 2.78 | **47.68** | 2.35 |

# 6 Analysis and Ablations

## 6.1 Same/different task classification and task embedding visualization

Capturing task nature is the prerequisite for task-specific adaptation conditioned on task embeddings. We evaluate the ability of gradients as task representations through a toy experiment and visualization. The toy experiment is a binary classification task about predicting whether two few-shot datasets are sampled from the same task or not. We name this task as "same/different task classification." For each pair of few-shot datasets, we calculate the ProtoNet loss and its gradients using the base model on the two datasets, respectively. We then feed the gradients on the two datasets into a single linear layer, respectively. Prediction is given by the cosine similarity between the two representations after linear transformation and the binary cross entropy loss is used for training.

We train the linear model on few-shot dataset pairs sampled from our meta-training datasets, and test it on dataset pairs sampled from our meta-testing datasets. Figure 2 shows the AUC curves on the testing set. Although the model is simple, it can predict same/different labels reasonably well on unseen tasks during training. With more shots, the gradient becomes more reliable, thus the model achieves better performance, resulting in an AUC score of 0.84 with 16 shots.

The results of the same/different task classification experiment confirm that gradients can be used as features to capture task nature and distinguish between tasks. To further evaluate the quality of the task embeddings learned by our model, we visualize the learned task embeddings in 2D space. We find the task embeddings form good quality clusters according to task nature at higher layers. More details can be found in Appendix E.

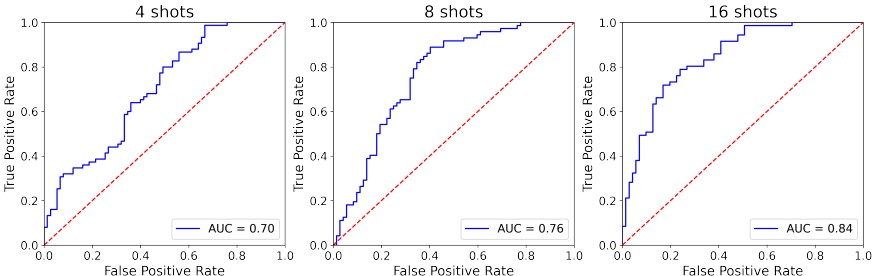

Figure 2: Results on the same/different experiments. Each column shows the results of classifying pairs of dataset with a certain number of shots.

## 6.2 Ablation Study

We conduct ablation studies to justify our decision choices. We report the average accuracy on meta-testing datasets of different model variants in Table 2. Results show that our approach achieves the best average accuracy among all model variants, resulting in average accuracy of 45.99%. The full results are included in Appendix F.

Table 2: Ablation results.

| Model | Mean Acc. |
|---|---|
| **Grad2Task w/ Gradients** | **45.99** |
| ProtoNet Longer Training | 45.10 |
| Grad2Task w/ X | 45.66 |
| Grad2Task w/ X&Y | 45.16 |
| Grad2Task Adapt All | 44.57 |
| Grad2Task w/ Pretrained TaskEmb | 45.68 |
| Hypernetwork | 44.79 |

**Does the adaptation really help?** Starting from the base model trained after the first stage, we compare our approach with just training the base model using the ProtoNet loss for the same number of steps. Shown as "ProtoNet Longer Training" in Table 2, the performance of this approach is worse than our proposed approach. This justifies that we can do better by adapting the base model than training it for more steps.

**How are gradients compared with other task representations?** We compare our approach with other CNP-based approaches with different methods of task embedding. Similar to Requeima et al. [37], we use the average embeddings of the training sequences as representations for a task, and keep the other model architecture unchanged. We also consider using the embeddings of both input sequences and labels for task representation. We treat labels as normal text instead of discrete values and encode them using the BERT model. The label embedding and average input embeddings are concatenated as the task representation. Shown as "Grad2Task w/ X" and "Grad2Task w/ X&Y", respectively, in Table 2, both of these two approach underperform our gradient-based method while "Grad2Task w/ X" performs closely with our method. Note that we can also combine those methods for task representation, which we leave for future work.

**Adapt all or just "[CLS]"?** The only difference between this model and ours is whether we apply the generated adaptation parameters only on the hidden representation of the "[CLS]" token or on all tokens in each sequence. Shown as "Grad2Task Adapt All" in Table 2, this model performs worse, resulting in 44.57% on average.

**Generating model parameters or adaptation parameters?** We also experimented with a hyper-network [20]-based approach. Conditioned on the task representation, we use a neural network (hypernetwork) to output the parameters of bottleneck adapters directly, instead of freezing the bottleneck adapters and only generating scaling/shifting parameters, as in our proposed model. The model with a hypernetwork has higher flexibility for task adaptation, but the size of the hypernework is large, as it maps high-dimensional gradient information to high-dimensional model parameters, which brings challenges for optimization and suceptibility for overfitting. As shown in Table 2, it achieves accuracy of 44.79% on average and is worse than our proposed approach.

**Pretraining task embedding?** For this model, we first pretrain the task embedding model for the same/different tasks, then freeze and combine it with other components following our proposed approach. We observe that it has close but slightly worse performance (45.68%) than Grad2Task without task embedding pretraining. We hypothesize the task embedding trained with the same/different tasks is not good enough. We expect this model will benefit from pretraining the task embedding

model on a set of datasets with higher diversity and training the task embedding model using more advanced metric-learning approaches, which we leave for future work.

# 7 Discussion and Conclusion

In this paper, we propose a novel model-based meta-learning approach for few-shot text classification and show the feasibility of using gradient information for task conditioning. Our approach is explicitly designed for learning from diverse text classification tasks how to adapt a base model conditioned on different tasks. We use pretrained BERT with bottleneck adapters as the base model and modulate the adapters conditioned on task representations based on gradient information.

Our work is an inaugural exploration of using gradient-based task representations for meta-learning. It has several limitations. First, the way we use neural networks to encode high-dimensional data significantly increases the number of parameters to train. For future work, we will explore more efficient ways for FIM calculation and better ways to handle high-dimensional gradient information. Second, we only focus on text sequence classification tasks in this paper. There are various types of NLP tasks to explore, such as question answering and sequence labeling tasks, which present many distinct challenges. Also, we only consider transferring between text classification tasks. As shown in Vu et al. [50], positive transfer can happen among different types of NLP tasks. Thus, we intend to explore meta-learning or transfer learning approaches to learning from different types of tasks.

**Ethical Concerns.** Our work advances the field of few-shot text classification. While not having any direct negative societal impact, one can imagine that a few-shot text classifier is abused by a malicious user. For example, classifying trendy keywords on the internet naturally falls into the application area of our method as example phrases containing the new keyword can be limited in number. Toxic language is known to be a pervasive problem in the Internet [3], and a malicious user could use a good few-shot text classifier to further perpetuate these problems (e.g., by classifying toxic text as benign and fooling the end user). On the other hand, the same system can be used to promote fairness.

# Acknowledgments and Disclosure of Funding

We would like to thank reviewers and ACs for constructive feedback and discussion. Resources used in preparing this research were provided, in part, by the Province of Ontario, the Government of Canada through CIFAR, and companies sponsoring the Vector Institute (`www.vectorinstitute.ai/#partners`). JW is supported by the RBC Graduate Fellowship. FR and MB are CIFAR AI Chairs.

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
