# Appendix

## A  Notations

In Table A1, we list the notions that are used throughout the paper.

Table A1: List of notations.

| Symbol | Description |
|---|---|
| $x$ | Input features. |
| $y$ | Output label. |
| $\mathcal{S}$ | A support set. |
| $\mathcal{Q}$ | A query set. |
| $\mathcal{S}^c$ | The subset of examples in $S$ that belong to class $c$: $\mathcal{S}^c = \{(x_i, y_i) \mid (x_i, y_i) \in \mathcal{S}, y_i = c\}$ |
| $f$ | The base model. We use a pretrained BERT model, insert bottleneck adapters into each transformer layer and add a linear output layer. |
| $\theta$ | Parameters of pretrained BERT model. |
| $\alpha$ | Parameters of the bottleneck adapters. |
| $\omega$ | Parameters of the linear output layer. |
| $\Theta$ | $\Theta = \{\theta, \alpha, \omega\}$. |
| $\mu^c$ | The centroid of $\mathcal{S}^c$ which is the average embedding of the examples in $\mathcal{S}^c$. |
| $\hat{p}(x)$ | The empirical distribution over $x$ specified by a dataset. |
| $\nabla_\Theta$ | The gradients with regard to $\Theta$. |
| $F_\Theta$ | The Fisher Information Matrix of $\Theta$. |
| $d$ | The task embedding network. |
| $\phi$ | Parameters of the task embedding network. |
| $a$ | The adaptation network. |
| $\psi$ | Parameters of the adaptation network. |
| $\gamma$ | Scaling parameters for adaptation. |
| $\beta$ | Shifting parameters for adaptation. |

## B  ProtoNet Training

Algorithm A1 shows the pseudocode for training the base model during the first stage. The training procedure follows Snell et al. [40] broadly. Different with regular episodic training where episodes are sampled from a single large dataset, we have multiple meta-training datasets to sample episodes from. Accordingly, we first sample a meta-training dataset with probability proportional to the square root of its size and then sample an episode from that dataset. We repeat this process to generate episodes for meta-training. We also make sure each episode contains equal number of examples for all classes of the dataset it is sampled from.

## C  Experiment Datasets

Table A2 shows the datasets we use for meta-training, which are the same with the datasets used by Bansal et al. [4] except for SST-2: Bansal et al. [4] used SST-2 as an entity typing task while we used it as a sentence-level sentiment classification task. We use the average performance on the validation sets of all meta-training tasks for hyperparameter searching and early stopping.

Table A3 lists the datasets we use for meta-testing. We consider 4-shot, 8-shot and 16-shot for each task. For each task with a certain number of shots, we train on 10 different training sets and test on the full testing set, and report the mean and standard deviation of the testing performance over the 10 runs. We reuse most of the meta-testing datasets used by Bansal et al. [4] but remove all the datasets of which the average number of words per input sentence is larger than 100. In addition, we add three new datasets to increase the diversity of the meta-testing tasks.

**Algorithm A1:** Training a prototypical network with BERT and bottleneck adapters as encoder.

**Input** : $T$: A set of text classification tasks for training. $\mathcal{D}_i$: The labeled dataset of each task $i$.
$\theta$: Parameters of pretrained BERT.

**Output :** $\alpha$: Parameters of the bottleneck layers. $\omega$: Parameters of the linear layer after BERT.

1 **while** *not converged* **do**
2     Sample $M$ episodes $\{t_i\}_{i=1}^M$, each episode $t_i = (\mathcal{S}_i, \mathcal{Q}_i), \mathcal{S}_i \subseteq \mathcal{D}_i, \mathcal{Q}_i \subseteq \mathcal{D}_i$
3     **for** $i \leftarrow 1$ **to** $M$ **do**
4        $\mu_c = \frac{1}{|\mathcal{S}^c|}\Sigma_{x_j \in \mathcal{S}^c} f(x_j; \theta, \alpha, \omega), S^c = \{(x_j, y_j)\,|\,(x_j, y_j) \in \mathcal{S}_i, y_j = c\}$    `// build`
                                                 `// prototype of each class`
5        $p(y = c|x_j) = \text{softmax}(\cos(f(x_j; \theta, \alpha, \omega), \mu_c)), (x_j, y_j) \in \mathcal{Q}_i$
6        $\ell_i = -\sum_{x_j \in \mathcal{Q}_i} \log p(y = y_j|x_j), (x_j, y_j) \in \mathcal{Q}_i$    `// calcualte loss on the`
                                                 `// query set`
7     **end**
8     $\{\alpha, \omega\} \leftarrow \{\alpha, \omega\} - lr \cdot \nabla_{\{\alpha, \omega\}} \frac{1}{M} \sum_{i=1}^M \ell_i$
9 **end**

Table A2: Meta-training datasets.

| Dataset | Task | Labels | #Training | #Validation |
|---|---|---|---|---|
| MRPC [15] | Paraphase | "paraphase", "not paraphase" | 3668 | 409 |
| QQP [21] | detection | "paraphase", "not paraphase" | 363846 | 40430 |
| QNLI [51] | | "entailment", "not entailment" | 104743 | 5463 |
| RTE [13] | NLI | "entailment", "not entailment" | 2490 | 277 |
| SNLI [10] | | "contradiction", "entailment", "neutral" | 549367 | 9824 |
| MNLI [52] | | "contradiction", "entailment", "neutral" | 392702 | 19647 |
| SST-2 [41] | Movie review classification | "negative", "positive" | 67349 | 872 |

Following Bansal et al. [4], we use several text classification datasets from crowdflower[2], including *airline*, *disaster*, *emotion*, *political_audience*, *political_bias* and *political_message*. We also use product rating dataset from the Amazon Reviews dataset [8] but only keep *rating_kitchen*. We add three different datasets to increase the diversity of the meta-testing tasks. We use a subset of the Yelp dataset [3], named *yelp* in Table A3. We randomly sample 2000 examples for each class as the testing set. For training, similar with other tasks, we randomly sample 10 training sets for each number of shots. SNIPS [11] is a commonly used benchmarking dataset for intent detection and slot filling tasks. We use the full test set of SNIPS and randomly sample few-shot datasets from its training set. The HuffPost dataset [26] is about classifying categories of news posted on HuffPost[4]. The full dataset contains 41 categories, from which we randomly select 10 categories and 400 examples for each category as the testing set.

We only considered the 12 datasets in the main table of [4] and removed the two entity typing tasks, because they are phrase-level classification tasks while we only focus on sentence-level classification tasks. And we removed the other 3 tasks because they contain many sentences that exceed the max sequence length of 128. Note that we use 128 as the max length for fair comparison with [6], since it had the same restriction.

# D   Experiment details

## D.1   Comparing our Leopard implementation with [4]

We compare the results of our implementation of ProtoNet (with BERT as encoder) and Leopard with the results reported by Bansal et al. [4]. Note that here we use the same meta-testing datasets as Bansal et al. [4] without removing the datasets that contain many long sentences, namely, *rating_dvd*,

---

[2]https://www.figure-eight.com/data-for-everyone/
[3]https://www.yelp.com/dataset
[4]https://www.huffpost.com/

Table A3: Meta-testing datasets. Newly added datasets are marked with "*".

| Dataset | Task | #Test Size | Labels |
|---|---|---|---|
| airline | Sentiment classification on tweets about airline | 7319 | "neutral", "negative", "positive" |
| rating_kitchen | Product rating classification on Amazon | 7379 | "4", "2", "5" |
| disaster | Classifying whether tweets are relevant to disasters | 5430 | "not relevant", "relevant" |
| emotion | Emotion classification | 20000 | "enthusiasm", "love", "hate", "neutral", "worry", "anger", "fun", "happiness", "boredom", "sadness", "surprise", "empty", "relief" |
| political_audience | Classifying the | 996 | "national", "constituency" |
| political_bias | audience/bias/message | 1287 | "partisan", "neutral" |
| political_message | of social media messages from politicians | 428 | "personal", "policy", "support", "media", "attack", "other", "information", "constituency", "mobilization" |
| snips* | Intent detection | 700 | "play music", "add to playlist", "rate book", "search screening event", "book restaurant", "get weather", "search creative work" |
| huffpost_10* | Category classification on news headlines from HuffPost | 4000 | "politics", "entertainment", "travel", "wellness", etc. |
| yelp* | Business rating classification on Yelp | 10000 | "1", "2", "3", "4", "5" |

*rating_electronics* and *rating_books*, and without adding new tasks for fair comparison. Results are shown in Table A4. The average accuracy of our Leopard implementation over all tasks is 47.59%, which is close to the average accuracy reported by Bansal et al. [4] (48.22%). However, the average accuracy of our ProtoNet implementation (51.22%) is much better than the performance reported by Bansal et al. [4] (42.36%), and is even better than Leopard. See Table A4 for detailed results on each task.

## D.2 Implementation details

Our codes are publicly available on https://github.com/jixuan-wang/Grad2Task. During meta-training, after sampling episodes with roughly the same number of examples as the total number of examples in the meta-training datasets, we refer this as one epoch. For all models, we train them on the meta-training datasets for 5 epochs and report results of the models with the best average performance on the validation sets of all meta-training tasks. To calculate the ProtoNet loss, we tried both Euclidean distance and cosine distance as the distance metric, and found that Euclidean distance worked better and so used it for all experiments. The linear layer on top of BERT has the size of 256. The task embedding network is a 2 layers GRU model, of which the input size is 24567 (i.e., the parameter size in each bottleneck adapter) and the output size is task embedding size (we used 100). The adaptation networks are single layer MLPs.

We use dropout rate [43] of 0.1 for all models. We choose learning rate for each model from $\{1e-5, 2e-5, 5e-5, 1e-4\}$ based on the validation performance. We use the Adam algorithm [23] for optimization. We perform one optimization step after seeing 4 episodes. Note that our approach does not require tuning any hyperparameters like the number of steps in inner loop as in MAML based approches [4]. Each model is trained on 2 NVIDIA Tesla P100 GPUs.

Parameter size of the BERT$_{\text{BASE}}$ model is 110M but all of its parameters are kept fixed in our method. We insert 24 bottleneck adapters into the BERT$_{\text{BASE}}$ model, which consume 7M parameters in total and are only trained in the first stage. The task embedding network contains 7M parameters. The adaptation network contains 1.3M parameters for each bottleneck adapter.

Table A4: Comparison between [4] and our implementation. "#" refers to number of shots. "Leopard ProtoNet" refers to the ProtoNet with BERT as encoder implemented by [4]. "Leopard" refers to the MAML-based approach proposed and implemented by [4]. "Our ProtoNet" refers to the ProtoNet with BERT as encoder implemented by us. "Our Leopard" refers to the Leopard model implemented by us. Note in this table, we **did not** remove the datasets containing very long sentences, namely, *rating_dvd*, *rating_electronics* and *rating_books*.

| # | model | Leopard ProtoNet | Leopard | Our ProtoNet | Our Leopard |
|---|---|---|---|---|---|
| | airline | 40.27 ± 8.19 | 54.95 ± 11.81 | **65.39 ± 12.73** | 48.21 ± 17.99 |
| | disaster | 50.87 ± 1.12 | 51.45 ± 4.25 | **54.01 ± 2.90** | 51.32 ± 4.11 |
| | emotion | 9.18 ± 3.14 | 11.71 ± 2.16 | **11.69 ± 1.87** | 11.27 ± 3.92 |
| | political_audience | 51.47 ± 3.68 | 52.60 ± 3.51 | 52.77 ± 5.86 | **53.54 ± 4.15** |
| | political_bias | 56.33 ± 4.37 | **60.49 ± 6.66** | 58.26 ± 10.42 | 58.08 ± 9.19 |
| 4 | political_message | 14.22 ± 1.25 | 15.69 ± 1.57 | **17.82 ± 1.33** | 16.82 ± 1.79 |
| | rating_books | 48.44 ± 7.43 | 54.92 ± 6.18 | **60.12 ± 8.05** | 56.94 ± 10.43 |
| | rating_dvd | 47.73 ± 6.20 | 49.76 ± 9.80 | **56.95 ± 10.17** | 44.68 ± 11.29 |
| | rating_electronics | 37.40 ± 3.72 | 51.71 ± 7.20 | **57.32 ± 7.61** | 51.61 ± 8.83 |
| | rating_kitchen | 44.72 ± 9.13 | 50.21 ± 9.63 | **58.47 ± 11.12** | 48.77 ± 12.44 |
| | Average | 40.06 ± 4.82 | 45.35 ± 6.28 | **49.28 ± 7.21** | 44.12 ± 8.41 |
| | airline | 51.16 ± 7.60 | 61.44 ± 3.90 | **69.14 ± 4.84** | 65.68 ± 11.79 |
| | disaster | 51.30 ± 2.30 | **55.96 ± 3.58** | 54.48 ± 3.17 | 50.37 ± 3.31 |
| | emotion | 11.18 ± 2.95 | 12.90 ± 1.63 | **13.10 ± 2.64** | 13.03 ± 5.99 |
| | political_audience | 51.83 ± 3.77 | 54.31 ± 3.95 | **55.17 ± 4.28** | 52.15 ± 5.57 |
| | political_bias | 58.87 ± 3.79 | 61.74 ± 6.73 | **63.22 ± 1.96** | 62.69 ± 1.08 |
| 8 | political_message | 15.67 ± 1.96 | 18.02 ± 2.32 | **20.40 ± 1.12** | 17.38 ± 1.92 |
| | rating_books | 52.13 ± 4.79 | 59.16 ± 4.13 | 62.59 ± 8.11 | **63.13 ± 8.08** |
| | rating_dvd | 47.11 ± 4.00 | 53.28 ± 4.66 | **59.18 ± 7.00** | 53.42 ± 9.54 |
| | rating_electronics | 43.64 ± 7.31 | 54.78 ± 6.48 | **61.57 ± 2.94** | 58.43 ± 3.08 |
| | rating_kitchen | 46.03 ± 8.57 | 53.72 ± 10.31 | **57.08 ± 11.54** | 53.19 ± 12.05 |
| | Average | 42.89 ± 4.70 | 48.53 ± 4.77 | **51.59 ± 4.76** | 48.95 ± 6.24 |
| | airline | 48.73 ± 6.79 | 62.15 ± 5.56 | **71.06 ± 1.60** | 67.04 ± 8.06 |
| | disaster | 52.76 ± 2.92 | **61.32 ± 2.83** | 55.30 ± 2.68 | 50.37 ± 4.27 |
| | emotion | 12.32 ± 3.73 | **13.38 ± 2.20** | 12.81 ± 1.21 | 10.30 ± 2.86 |
| | political_audience | 53.53 ± 3.25 | **57.71 ± 3.52** | 56.16 ± 2.81 | 54.94 ± 2.34 |
| | political_bias | 57.01 ± 4.44 | **65.08 ± 2.14** | 61.98 ± 6.89 | 61.38 ± 5.03 |
| 16 | political_message | 16.49 ± 1.96 | 18.07 ± 2.41 | **21.36 ± 0.86** | 18.22 ± 1.88 |
| | rating_books | 57.28 ± 4.57 | 61.02 ± 4.19 | **65.82 ± 4.65** | 63.98 ± 9.32 |
| | rating_dvd | 48.39 ± 3.74 | 53.52 ± 4.77 | **61.86 ± 1.89** | 55.27 ± 8.91 |
| | rating_electronics | 44.83 ± 5.96 | 58.69 ± 2.41 | **60.49 ± 4.86** | 58.05 ± 3.49 |
| | rating_kitchen | 49.85 ± 9.31 | 57.00 ± 8.69 | **61.00 ± 9.17** | 57.34 ± 10.82 |
| | Average | 44.12 ± 4.67 | 50.79 ± 3.87 | **52.78 ± 3.66** | 49.69 ± 5.70 |

Training PN-BERT and PN-BN took 9 and 7 hours, respectively, on two Tesla P100 GPUs. In the second stage of our method, the task embedding network and adaptation network are trained together for 5 epochs on the meta-training datasets, which took 4 hours on two Tesla P100 GPUs.

# E    Visualization of task embeddings

We visualize the task embeddings learned by the task embedding network in Figure A1. We show the per-layer task embeddings of the meta-training tasks after being mapped into the 2D space by t-SNE [47]. Each point in Figure A1 is corresponding to an episode or task sampled from a meta-training dataset. For each episode, we first calculate the raw gradient information and then feed it into the RNN-based task embedding network, which outputs the per-layer task embeddings. Note that for the 12-layer BERT$_{BASE}$ model we inserted two adapters into each transformer layer, so there are 24 task embeddings in total for each episode. Here we only visualize the first task embedding at each transformer layer, resulting in 12 task embeddings for each episode. From Figure A1 we observe that the task embeddings form better clusters at higher layers. At the last layer, the only movie review classification task (SST-2) is separated clearly from other tasks, multiple NLI tasks are mixed together in the top center, and the paraphrase detection tasks QQP and MRPC spread in the center.

# F    Ablation results details

Details of the ablation study results are shown in Table A5. First, all variants under our framework achieve better performance than previous work [4] and other baselines, i.e., ProtoNet [40]-based and Hypernet [19]-based approaches. Second, overall, the model with gradients as task representation

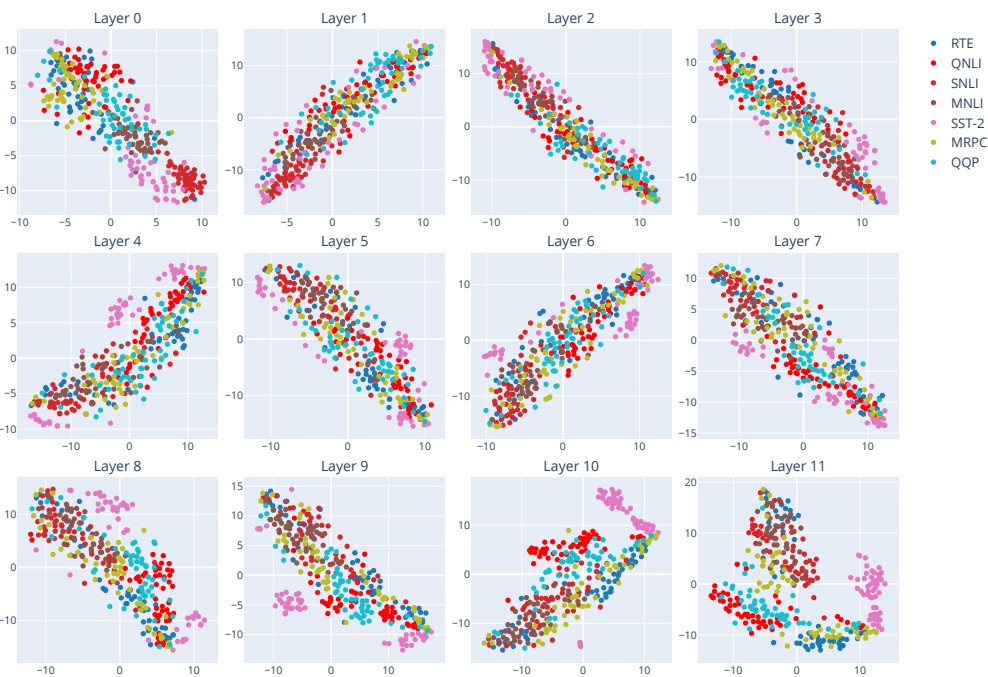

Figure A1: T-SNE visualization of the learned task embeddings. Each sub-figure is the task embeddings at a certain layer, e.g., "Layer 11" refers to the task embeddings at the last transformer layer of $BERT_{BASE}$. Each point is an episode or task sampled from a dataset. Each color is corresponding to a dataset.

(shown as "Grad2Task w/ Gradients") performs the best: for the ten meta-testing tasks being considered, it achieves the best performance among all the variants for five 4-shot tasks, four 8-shot tasks and four 16-shot tasks.

# G    Additional results

## G.1    Results on all datasets

We did not report results on the datasets used by [4] for domain adaptation and ablations, since we focus on evaluation generalization to tasks with different structure. However, we did run evaluation on all datasets of [4] (except entity typing datasets since they are phrase-level classification tasks). Table A6 shows the average accuracy under each shot, demonstrating that our conclusion holds.

## G.2    Comparison with other fine-tuning baselines

The main disadvantage of fine-tuning approaches is that they require costly retraining on new tasks and hand tuning and are more vulnerable to overfitting on few-shot problems. To illustrate this we compare different fine-tuning approaches in the table below (FT: fine-tuning, BN: bottleneck adapters, PN: ProtoNet, *: numbers reported in [4]). We show the average accuracy on a subset of tasks from [4] with 4, 8 and 16 shots. For example, FT-PN-BN refers to only fine-tuning the BN parameters of a trained ProtoNet. Results show all fine-tuning approaches perform much worse on 4-shot problems than our method, while some of them start to catch up with more shots, i.e., 16-shot.

Table A5: Detailed results of the ablation study. "#" refers to number of shots. We show both the mean and standard deviation of the accuracy over 10 runs. For example, in "66.78 ± 6.27", "66.78" refers to the average accuracy and "6.27" refers to the standard deviation of accuracy. We further average the results of each model for each number of shots, shown as the rows in grey color. All models shown in this table are trained by starting from the same base model after the first training stage. "PN Longer Training" refers to training the base model for more iterations without any adaptation. "Grad2Task X" is similar with our proposed model but uses average input encoding as task representation. "Grad2Task X&Y" is similar with our proposed model but uses average input encoding and textual label encoding as task representation. "Grad2Task Adapt All" refers to our proposed model but adapting the hidden representation of all input tokens instead of just the "[CLS]" token. "Grad2Task w/ Pretrained Emb" refers to our proposed model but using task embedding model pretrained on the same/different task. "Hypernet" is similar with our proposed approach but generating whole parameters for the bottleneck adapters instead of adaptation parameters. "Grad2Task w/ Gradients" refers to our proposed approach.

| # | Model: | PN Longer Training | Grad2Task X | Grad2Task X&Y | Grad2Task Adapt All | Grad2Task w/ Pretrained Emb | Hypernet | Grad2Task w/ Gradients |
|---|---|---|---|---|---|---|---|---|
| | airline | 66.78 ± 6.27 | 66.58 ± 11.92 | 66.88 ± 11.55 | 66.83 ± 12.17 | 67.76 ± 10.48 | 62.99 ± 7.54 | **70.64 ± 3.95** |
| | disaster | 53.46 ± 3.64 | 54.97 ± 6.83 | 53.54 ± 4.34 | 54.22 ± 5.93 | 54.83 ± 6.19 | 54.84 ± 5.69 | **55.43 ± 5.89** |
| | emotion | 12.64 ± 1.98 | **13.27 ± 1.90** | 12.80 ± 1.60 | 13.21 ± 2.27 | 12.80 ± 1.64 | 13.16 ± 1.17 | 12.76 ± 1.35 |
| | political_audience | 52.78 ± 5.57 | **52.79 ± 6.99** | 51.32 ± 6.15 | 53.01 ± 7.00 | 52.27 ± 6.15 | 50.59 ± 4.80 | 51.28 ± 5.74 |
| | political_bias | 63.52 ± 1.94 | **62.24 ± 6.50** | 59.63 ± 7.79 | 61.51 ± 7.07 | 60.30 ± 7.04 | 60.35 ± 6.72 | 58.74 ± 9.43 |
| 4 | political_message | 20.69 ± 1.26 | 20.87 ± 1.36 | 20.22 ± 1.81 | 19.69 ± 1.53 | 20.31 ± 1.65 | 17.18 ± 1.77 | **21.13 ± 1.97** |
| | rating_kitchen | 55.29 ± 10.29 | 56.27 ± 9.19 | 57.22 ± 10.25 | 56.69 ± 10.75 | 56.88 ± 9.71 | 56.78 ± 8.12 | **57.09 ± 9.74** |
| | huffpost_10 | 17.59 ± 2.69 | 17.01 ± 1.80 | 17.14 ± 1.77 | 16.94 ± 1.88 | 17.67 ± 2.27 | 17.00 ± 2.63 | **18.50 ± 2.00** |
| | snips | 47.77 ± 4.08 | 50.16 ± 2.89 | 47.46 ± 3.88 | 43.36 ± 2.39 | 49.09 ± 4.11 | 49.76 ± 4.67 | **52.51 ± 2.68** |
| | yelp | 41.68 ± 3.07 | **43.04 ± 2.63** | 41.65 ± 2.90 | 42.45 ± 3.25 | 42.50 ± 3.48 | 42.20 ± 3.15 | 43.00 ± 3.55 |
| | **Average** | 43.22 ± 4.08 | 43.72 ± 5.20 | 42.79 ± 5.20 | 42.79 ± 5.42 | 43.44 ± 5.27 | 42.48 ± 4.63 | **44.11 ± 4.63** |
| | airline | 70.27 ± 2.11 | 71.86 ± 3.52 | 71.67 ± 3.31 | 71.49 ± 2.23 | **72.51 ± 2.25** | 68.44 ± 2.59 | 72.04 ± 2.58 |
| | disaster | 54.75 ± 3.88 | 56.71 ± 6.55 | 56.59 ± 3.14 | 55.86 ± 3.99 | 55.34 ± 3.40 | 55.79 ± 4.16 | **57.49 ± 5.36** |
| | emotion | 13.62 ± 1.79 | 14.14 ± 2.20 | 13.92 ± 1.60 | 14.15 ± 2.38 | **14.46 ± 2.08** | 14.33 ± 1.44 | 13.99 ± 1.90 |
| | political_audience | 53.46 ± 5.05 | 53.65 ± 6.44 | 53.55 ± 6.19 | **54.56 ± 5.62** | 52.97 ± 5.79 | 52.39 ± 4.69 | 52.60 ± 5.55 |
| | political_bias | 64.69 ± 0.73 | **65.07 ± 0.76** | 64.25 ± 0.96 | 64.32 ± 0.44 | 64.90 ± 1.12 | 63.68 ± 1.43 | 64.06 ± 1.12 |
| 8 | political_message | **21.76 ± 0.89** | 21.71 ± 1.77 | 21.13 ± 1.81 | 21.29 ± 1.03 | 21.04 ± 1.76 | 20.18 ± 2.05 | 21.31 ± 1.16 |
| | rating_kitchen | 56.68 ± 10.99 | 57.50 ± 10.17 | 58.11 ± 9.77 | 56.62 ± 10.61 | 57.72 ± 10.41 | 55.46 ± 10.99 | **58.35 ± 9.83** |
| | huffpost_10 | 19.81 ± 2.53 | 19.97 ± 1.47 | 20.31 ± 1.74 | 19.07 ± 1.75 | 20.97 ± 1.54 | 18.93 ± 1.62 | **21.12 ± 1.69** |
| | snips | 54.27 ± 3.11 | 53.81 ± 2.63 | 52.56 ± 2.23 | 48.49 ± 4.41 | 53.83 ± 2.30 | 56.41 ± 3.47 | **57.19 ± 2.77** |
| | yelp | 43.26 ± 1.85 | **45.15 ± 1.61** | 44.20 ± 1.69 | 43.73 ± 1.73 | 44.26 ± 1.98 | 42.90 ± 3.19 | 43.66 ± 1.65 |
| | **Average** | 45.26 ± 3.29 | 45.96 ± 3.71 | 45.63 ± 3.24 | 44.96 ± 3.42 | 45.80 ± 3.26 | 44.85 ± 3.56 | **46.18 ± 3.36** |
| | airline | 70.20 ± 1.62 | 72.25 ± 1.94 | 72.09 ± 1.93 | 71.64 ± 1.63 | **72.79 ± 1.58** | 67.87 ± 1.62 | 72.30 ± 1.75 |
| | disaster | 57.05 ± 4.46 | 57.46 ± 2.47 | 58.45 ± 2.92 | 56.73 ± 3.37 | 58.83 ± 5.32 | 58.94 ± 3.56 | **59.63 ± 3.11** |
| | emotion | 14.28 ± 1.46 | 14.02 ± 0.93 | 14.04 ± 1.29 | 13.91 ± 0.78 | 14.23 ± 1.32 | **14.54 ± 0.90** | 13.72 ± 1.24 |
| | political_audience | 56.86 ± 3.01 | **57.79 ± 4.02** | 56.71 ± 3.62 | 56.98 ± 4.15 | 56.99 ± 2.44 | 55.79 ± 2.72 | 55.46 ± 3.34 |
| | political_bias | 64.67 ± 0.72 | **65.36 ± 0.77** | 64.06 ± 0.53 | 64.21 ± 0.18 | 64.29 ± 2.88 | 64.33 ± 0.60 | 63.83 ± 0.74 |
| 16 | political_message | 23.44 ± 0.91 | 23.64 ± 1.26 | 23.00 ± 1.19 | 22.31 ± 1.23 | **23.89 ± 1.11** | 22.29 ± 2.32 | 22.22 ± 1.20 |
| | rating_kitchen | 59.64 ± 9.14 | 61.03 ± 6.64 | 60.72 ± 6.38 | 59.35 ± 8.78 | 61.68 ± 6.81 | 59.48 ± 7.01 | **61.72 ± 6.38** |
| | huffpost_10 | 21.96 ± 1.30 | 21.45 ± 1.87 | 21.98 ± 1.78 | 20.48 ± 1.73 | 22.56 ± 1.27 | 21.34 ± 2.19 | **23.57 ± 1.76** |
| | snips | 56.21 ± 2.40 | 55.65 ± 1.55 | 55.16 ± 2.37 | 49.89 ± 2.96 | 57.19 ± 2.08 | 61.81 ± 1.45 | **59.47 ± 1.91** |
| | yelp | 43.80 ± 2.17 | 44.43 ± 2.90 | 44.53 ± 2.23 | 44.25 ± 2.25 | **45.39 ± 1.35** | 43.85 ± 2.14 | 44.87 ± 2.09 |
| | **Average** | 46.81 ± 2.72 | 47.31 ± 2.44 | 47.07 ± 2.42 | 45.97 ± 2.71 | **47.78 ± 2.62** | 47.02 ± 2.45 | 47.68 ± 2.35 |

Table A6: Average accuracy on all datasets used by Bansal et al. [4]. "*": results reported by [4].

| Model | K4 | K8 | K16 |
|---|---|---|---|
| Leopard* | 52.61 | 56.02 | 57.97 |
| PN-BN | 56.5 | 57.94 | 59.69 |
| Grad2Task | 57.05 | 58.37 | 59.7 |

Table A7: Comparison with other fine-tuning approaches. "*": results reported in [4].

| model | K4 | K8 | K16 |
|---|---|---|---|
| FT-BERT* | 37.79 | 37.05 | 46.28 |
| FT-BERT-BN | 37.29 | 36.96 | 37.92 |
| FT-PT-BN-FILM | 40.95 | 45.83 | 49.74 |
| FT-PT-BN | 41.68 | 46.33 | 49.27 |
| FT-PT | 41.52 | 47.24 | 49.85 |
| FT-BERT | 38.5 | 39.4 | 45.21 |
| PN | 45.49 | 47.51 | 48.52 |
| PN-BN | 45.98 | 47.64 | 49.63 |
| Grad2Task | 46.72 | 48.55 | 49.84 |