# OpenReview forum: "Grad2Task: Improved Few-shot Text Classification Using Gradients for Task Representation"
_NeurIPS.cc/2021/Conference — NeurIPS 2021 Poster_

### Official Review · Reviewer_Ywoc · 2021-07-06

**Rating:** 6
**Confidence:** 3

**Summary:**

  A new method for few-shot text classification

**Limitations And Societal Impact:**

Not convincing; issues raised have to do with text classification n in general, b not so much with the paper

**Main Review:**

Th paper has a good analytical presentation and a set of thorough experiments that demonstrate it is better on the selected datasets. It is not clear how many labeled instances are required, and at which threshold the method degrades.

**Time Spent Reviewing:**

0.33

---

> ### Author Response · Authors · 2021-08-10
> **Thank you for your questions. We added our answers.**
>
> Thank you for your time reviewing our paper. We answer your questions as follows:
>
> > It is not clear how many labeled instances are required, and at which threshold the method degrades.
>
> We investigate the performance of different methods with different numbers of labeled data points, or shots. There is no absolute performance metric to determine how many labeled instances are required, but in general, performance improves with more shots, and we show relative improvement of our method over other methods with different shots.
>
> > Issues raised have to do with text classification in general, but not so much with the paper
>
> The problem we focus on in this paper is few-shot text-classification. The method we designed is specifically for learning from a few labeled examples, not from standard large-scale datasets. Also, we tried to cover a broad range of text classification tasks to more thoroughly measure the performance of each method.

---

### Official Review · Reviewer_WPsh · 2021-07-07

**Rating:** 5
**Confidence:** 5

**Summary:**

The paper considers the problem of few-shot text classification. It builds on a bunch of prior works on meta-learning [1,2,3], Film adapters [4], task embeddings [5] and tries to combine them for the task of few-shot text classification. The key idea seems to be to use adapters in between transformer layers, compute task embeddings (using fisher Information matrix) for a task and use it to update Film parameters that are used to update the model. The approach is compared with a prior work [1] and prototypical networks on a subset of few-shot classification tasks selected from [1].


**Limitations And Societal Impact:**

There is an appropriate discussion.

**Main Review:**

I like the paper’s proposal to use task-embeddings to further modulate the encoder for few-shot learning. This is not a new concept and has been widely explored in meta-learning literature. The main contribution of the paper seems to be in its specific application to text classification models and the use of the task embeddings to generate the Film parameters.

Overall, I found the paper very hard to follow, there are numerous typos and the exposition made it really difficult to extract the key contribution, the motivations for the specific choices, and the key take-aways. The paper seems a lot like an amalgamation of many related concepts, which is not a bad thing but in the current version of the paper there is no clear motivation for the various choices. To provide just one instance, if the main motivation is to use task embeddings then there are many straight-forward ways to incorporate them in existing models, such as in the input to the encoder or the classifier, and carefully considering/evaluating such choices would have been useful in motivating the approach. Another choice made by the authors is to use only the values in FIM corresponding to the adapters. Why is it sufficient to only use these, is there any analysis on the quality of the task embedding when restricting to only the adapters?
Similarly, the motivation for using task embeddings to generate Film adapters is not clear. For instance, one can use Film adapters with regular fine-tuning on the few-shot set to learn “task-specific” Film adapters. Note that this baseline is not present in the paper. So how is including the task embedding (which also uses gradient information) helpful over fine-tuning? Motivation as well as empirical results in this direction are lacking.

Moreover, there are several concerns regarding experiments.
- First, as mentioned above, there are relevant baselines that need to be provided to understand the value of the proposed approach. These also include: ProtoNet with fine-tuning, ProtoNet with adapters fine-tuning, ProtoNet-BN with only Film fine-tuning. Currently, the main baselines for the paper are ProtoNet and ProtoNet-BN, none of which use gradient information like the proposed method. These baselines will help demonstrate the utility of task embeddings over fine-tuning.
- The authors use datasets from [1]. However, I see there are 17 datasets used in that paper and the authors picked 7 of them. I am not convinced that seq-length is an appropriate reason to drop 10 datasets, for instance, the authors can run the model with a larger sequence length during evaluation. Moreover, there are decisions like only using a subset of classes in one of the datasets (Huffpost), and why this is done is also not clear. As such, due to the use of non-standard datasets, I am not sure how to interpret the results.
- Ablations are carried on all the downstream tasks used for final evaluation, which makes me think the results are overfit to these tasks. I didn’t find these particularly insightful either. For example, ablations on these choices would have been more helpful: using/not-using adapters or Film adapters, hidden-size of adapters, fine-tuning adapters vs using task embeddings, alternatives to incorporate task embeddings.
- Some hyper-parameter choices are missing. For example in section 4.2, what are the hyper-parameters of the RNN d()? Similarly, what is the architecture and hyper-parameters of the MLPs a(.) in adaptation network?
- There is no analysis on what the task embeddings are capturing and why they are useful here. This would have helped motivate the paper quite a bit.

These concerns together make me hesitant about the value of the task-embedding approach presented here. The authors should consider a thorough re-writing of the paper to provide motivation and to explain their approach clearly, and consider the suggested baselines and improvements to experiments explained above.

[1] Learning to few-shot learn across diverse natural language classification tasks.
[2] Meta-dataset: A dataset of datasets for learning to learn from few examples.
[3] Fast and Flexible Multi-Task Classification Using Conditional Neural Adaptive Processes.
[4] Film: Visual reasoning with a 382 general conditioning layer.
[5] Task2vec: Task embedding for meta-learning.


Update post rebuttal

The authors provided a lengthy response with additional experiments. These partly address some concerns but raise a few more.
When using more tasks and relevant baselines as reported here, I can see that the margins of improvements are significantly less than what is reported in the paper. For ex, FT-PN-BN is competitive with the proposed method. Some of the new results are also very puzzling. Like why does prototypical networks after fine-tuning becomes much worse than without fine-tuning. Another related problem I hadn't noticed in the first review was the difference between prototypical and other fine-tuning baselines, this difference seems too large, contrary to previous literature, and warrants more exploration. Comparing some results in the rebuttal and in Table 1, it seems that most of the improvements for the proposed approach are coming from the 3 new introduced datasets. These results are also very surprising. For instance, what is the property of SNIPS dataset on which Leopard and PN suddenly perform at half the accuracy of Grad2Task (21, 28 vs 52)!

I have still raised my score in favor weak accept since the rebuttal did address some concerns. Authors should incorporate these results and the presentation improvements in the paper.

**Time Spent Reviewing:**

6

---

> ### Author Response · Authors · 2021-08-10
> **Thank you for valuable feedback. We clarified our motivation and contributions, and provided additional experimental results.**
>
> Thank you for your valuable comments. We agree that many of these suggestions will make our paper stronger, and we have addressed them below. We also believe that our motivation and major contributions may not have been clearly explained. In the new version (and in this response) we clarify our motivation and contributions and present additional experimental results to support our claims.
>
> **Motivation:** Our work is motivated by an important yet underexplored few-shot learning problem in the NLP domain: learning from a set of supervised tasks to better solve diverse few-shot learning problems, which is different from the conventional setup with homogenous task structures. This challenging problem requires more task-specific adaptation due to the distinct nature among tasks and resistance to overfitting due to its few-shot nature, which motivate many of our modeling choices.
>
> **Contributions:** We propose the use of gradients as features to represent tasks and to integrate these gradient features into a model-based meta-learning framework. We show that our method outperforms fine-tuning and other meta-learning approaches on a set of diverse tasks. Furthermore, we believe our work will inspire future work to investigate how to utilize the gradient information of pretrained models as additional features to solve downstream tasks.
>
> Please see the following for our answers to your other questions and additional experimental results:
>
> > Q1. Why not try other ways to incorporate task embeddings?
>
> FiLM conditioning is widely used and has been shown to be a very expressive mechanism on a wide range of tasks despite being parameter light [1,2,3], and achieves impressive performance on solving diverse tasks [2,3] which aligns well with our goal.
>
> The way in which we use task embeddings is motivated by [2] and is a compromise between flexibility, efficiency, and resistance to over-fitting. We considered three types of task conditioning: (1) using task embeddings as input, as in CNP [4], and mentioned by the reviewer; (2) generating adaptation parameters conditioned on the task embeddings, as in CNAP and our work; and (3) generating the parameters of the base model directly conditioned on the task embeddings, as in [5]. (1) is the least flexible while (3) is the most flexible.
>
> The reason we chose (2) over (1) is because, as pointed out in [2], (1) is designed for homogeneous tasks that share the same structure, which lacks the flexibility to handle heterogeneous tasks that we consider. The reason we chose (2) over (3) is because the high flexibility of (3) also makes it vulnerable to overfitting. We tried this approach and named it as “Hypernetwork” in our ablation study. As shown in Table 2, it performed worse than our approach.
>
> > Q2. Why only use bottleneck adapters’ gradients?
>
> (1) This simplifies the use of gradients for task representation. Using the gradients of the whole BERT model will result in an input dimension of 110M, which is too large to be fed into another network.
>
> (2) In Section 6.1, we showed that the task embeddings based on adapters’ gradients can be used to identify whether two new tasks are from the same dataset or not, which confirms the ability of the learned embedding to represent task nature.
>
> > Q3. What is the motivation of generating adaptation parameters instead of fine-tuning?
>
> We want to avoid fine-tuning when seeing new tasks because it needs costly retraining and hand tuning, which makes it less user-friendly. In contrast, our approach is amortized in the sense that it uses fixed functions to generate parameters directly, so that no gradient descent is needed on new tasks (note that during testing, we only use gradients as features, not for updating parameters).
>
> > Q4. Unfair comparison since main baselines do not use gradient information as the proposed method
>
> We would like to clarify that one of our main baselines is Leopard, which is a MAML-based approach and uses gradient descent for adaptation on new tasks. We show results of ProtoNet with further fine-tuning in Q5.
>
> > Q5. Lack of baselines: Why not just fine-tune the adapter or FiLM parameters?
>
> The main reason we avoid fine tuning is that it requires costly retraining on new tasks and hand tuning, as well as because it is more vulnerable to overfitting on few-shot problems.
>
> To illustrate this we compared different fine-tuning approaches mentioned by the reviewer in the table below (FT: fine-tuning, BN: bottleneck adapters, PN: ProtoNet, *: numbers reported by [6]). We show the average accuracy on a subset of tasks from [6] with 4, 8 and 16 shots (see Q6 for why we used a subset). For example, FT-PN-BN refers to only fine-tuning the BN parameters of a trained ProtoNet. Results show all fine-tuning approaches perform much worse on 4-shot problems than our method, while some of them start to catch up with more shots, i.e., 16-shot.
>
> | model                 |    K4 |    K8 |   K16 |
> |:-|-:|-:|-:|
> | FT-BERT*          | 37.79  | 37.05  | 46.28 |
> | FT-BERT-BN    | 37.29 | 36.96 | 37.92 |
> | FT-PN               | 41.52 | 47.24 | 49.85 |
> | FT-PN-BN         | 41.68 | 46.33 | 49.27 |
> | FT-PN-FILM      | 40.95 | 45.83 | 49.74 |
> | PN                     | 45.49 | 47.51 | 48.52 |
> | PN-BN               | 45.98 | 47.64 | 49.63 |
> | Ours                  | 46.72 | 48.55 | 49.84 |
>
> > Q6. Why only used 7 datasets from [1]?
>
> We only considered the 12 datasets in the main table of [6] and removed the two entity typing tasks, because they are phrase-level classification tasks while we only focus on sentence-level classification tasks. And we removed the other 3 tasks because they contain many sentences that exceed the max sequence length of 128. Note that we use 128 as the max length for fair comparison with [6], since it had the same restriction.
>
> We didn’t report results on the datasets used by [6] for domain adaptation and ablations, since we focus on evaluation generalization to tasks with different structure.
>
> However, we did run evaluation on all datasets of [6] (except entity typing datasets). The table below shows the average accuracy under each shot, demonstrating that our conclusion holds. We will include the details in our appendix.
>
> | model          |    K4 |    K8 |   K16 |
> |:-|-:|-:|-:|
> | Leopard*    | 52.61 | 56.02 | 57.97 |
> | PN-BN        | 56.5  | 57.94 | 59.69 |
> | Ours           | 57.05 | 58.37 | 59.7  |
>
> > Q7. Are the datasets trustful?
>
> There is no standard dataset because this problem is relatively new and has only been previously studied in [6]. However, we compared different models using exactly the same datasets for fair comparison, and the datasets are publicly available.
>
> > Q8. Concerns about the ablation
>
> **Overfitting issue:** Note that we didn’t use any dataset in the ablation for model selection, so there is no way to over-fit on those tasks.
>
> **No ablation on the size of the adapters:** As shown in [7], the performance of BERT with bottleneck adapters is consistent with respect to adapter size. In our case, we keep it as small as possible to limit the number of gradients to consider.
>
> See Q1 and Q5 which address your other concerns about the missing ablations.
>
> > Q9. Missing hyperparameters
>
> Thank you for pointing this out. The missing hyperparameters are as follows: d(.): 2 layers GRU, the input size is 24567 (i.e., the parameter size in each bottleneck adapter) and the output size is task emb size (we used 100). a(.): single layer MLP, input/output size is determined by the base model. All these are included in our code repository, and we will add missing details to our paper.
>
> > Q10. What do the task embeddings capture and why are they useful?
>
> As shown in Section 6.1, the gradient-based task embeddings can distinguish reasonably well whether two new tasks have the same structure or not, which shows that our task embeddings are capable of capturing the nature of the task.
>
> Capturing task nature is the prerequisite for task-specific adaptation conditioned on task embeddings. We empirically show the usefulness of task embeddings by showing the model with task-specific adaptation outperforms the model without adaptation, as in Table 1. We also performed qualitative analysis demonstrating that the learned task embeddings can cluster tasks according to their nature reasonably well, which we will include in our appendix.
>
> Others: \
> Why only use 10 classes of Huffpost: The full Huffpost dataset contains 41 classes which is too difficult to generalize for any meta-learning approaches, since our training task at most contains 3 classes.
>
> [1] Oreshkin, Boris N., Pau Rodriguez, and Alexandre Lacoste. TADAM: Task dependent adaptive metric for improved few-shot learning.  NeurIPS 2018.
>
> [2] Requeima, James, et al. Fast and flexible multi-task classification using conditional neural adaptive processes. NeurIPS 2019.
>
> [3] Triantafillou, Eleni, et al. Learning a Universal Template for Few-shot Dataset Generalization. ICML 2021.
>
> [4] Garnelo, Marta, et al. Conditional Neural Processes. ICML 2018.
>
> [5] Ha, David, Andrew Dai, and Quoc V. Le. HyperNetworks. ICLR 2017.
>
> [6] Bansal, Trapit, Rishikesh Jha, and Andrew McCallum. Learning to few-shot learn across diverse natural language classification tasks. COLING 2020.
>
> [7] Houlsby, Neil, et al. Parameter-Efficient Transfer Learning for NLP. ICML 2019.

---

### Official Review · Reviewer_RV6z · 2021-07-15

**Rating:** 7
**Confidence:** 4

**Summary:**

This paper proposed Grad2Task, a model for few-shot text classification tasks. Grad2Task first computes gradients when fine-tuning the base model on a support set; then the task embedding network decodes task representations using these gradients; finally the task representations go through MLP layers and output scaling and shifting parameters, which are plugged into the base model for task-specific adaptation.

The paper provides comprehensive evaluation on 10 tasks. Grad2Task outperforms strong baselines (Leopard, ProtoNet-BERT) in most settings. Analysis with a toy dataset shows the task representations are informative. Ablation study justified several design choices in Grad2Task.

**Limitations And Societal Impact:**

The authors provide detailed discussion on Grad2Task's limitations in Sec 7. I listed some of my concerns and suggestions in the main review.

**Main Review:**

Strength:
- Novelty. The work is the first to explore gradient-based task representations for meta-learning. In addition to combining recent efforts (e.g., Task2Vec, meta-learning), the authors also reduces parameter size using lightweight design (e.g., adapters, shifting and scaling parameters).
- Detailed ablation study that explains the design choices in detail. The failure cases also brings interesting observations. Future work may stem from these observations.
- Strong performance when compared to strong baselines.

Weakness:
- Notation are sometimes confusing. (1) In Eq.3 the function d is overloaded. The function outputs the task embedding in the first part. It outputs RNN hidden representation in the second part. (2) In line 7-8 in Algorithm 1, e_j is a new variable that seems to be newly-introduced. Please also provide hints for the reader to link e_j with p(y|x_j), e.g., refer the reader to Eq. 1.

Questions:
- Is there any specific reason that RNN is selected as the task embedding network?
- How long is the training process? How large is the final model? Some comparison in model size and training efficiency may be included.

Minor Suggestions:
- Eq. 7. Is the first stage done on the support set or the query set or both?
- Line 274: “w/ XY”, Table 2 “w/ X&Y”. Please keep consistent.
- I believe the right side of Fig 1 is actually zooming in the part of [e_l^{task}] —> [Bottleneck Adapter]. Please consider highlighting this information to make the figure more accessible.


**Time Spent Reviewing:**

3

---

> ### Author Response · Authors · 2021-08-10
> **Thank you for your comments/suggestions. We answered your questions and provided more training details.**
>
> We thank the reviewer for the valuable comments and suggestions. We address your questions below:
>
> > Reason to use RNN as the task embedding network?
>
> First of all, thank you for pointing this out as a question.  We definitely see why answering this question can further clarify our contributions. The reasons are as follows (also integrated into the manuscript): (1) As shown in [1], providing information to higher layers about how lower layers have adapted to the current task is important for task specific adaptation. Inspired by this work, we use a RNN to summarize the gradient information from the lower layers so that the higher layers can better adapt to the current tasks. (2) Using RNNs can enable parameter sharing across different layers. Concatenating gradients of all layers as input will result in a very large task embedding model.
>
> > Training time/Model size
>
> Parameter size of the BERT-base model is 110M but all of its parameters are kept fixed in our method. We insert 24 bottleneck adapters into the BERT model, which consume 7M parameters in total and are only trained in the first stage. The task embedding network contains 7M parameters. The adaptation network contains 1.3M parameters for each bottleneck adapter.
>
> In the first stage, a ProtoNet is trained for 5 epochs on the meta-training datasets. Training PN-BERT and PN-BN took ~9 and ~7 hours, respectively, on two Tesla P100 GPUs.
> In the second stage, the task embedding network and adaptation network are trained together for 5 epochs on the meta-training datasets, which took ~4 hours on two Tesla P100 GPUs.
>
> Our implementation of Leopard took ~11 hours when training for 5 epochs on two Tesla P100 GPUs.
>
> > Eq. 7. Is the first stage done on the support set or the query set or both?
>
> Both are used. Eq. 7 shows the loss computed on one episode with both the support and query set: using the support set to build prototypes, predicting on the query set and calculating the cross entropy loss.
>
> > Overloaded function $d$ in Eq. 3
>
> Function $d$ in Eq.3 refers to the RNN model which outputs both the hidden state and task embedding at each step.
>
> Others: \
> We will make it clear the relationship between $e_j$ and $p(y|x_j)$ in Alg. 1.
>
> Thank you very much for pointing out the typos and suggestions for the figure. We will carefully address them in the revision.
>
> [1] Requeima, James, et al. Fast and flexible multi-task classification using conditional neural adaptive processes. NeurIPS 2019.

---

> > ### Comment · Reviewer_RV6z · 2021-08-24
> > **Reply**
> >
> > Thank you for your response. Please add these discussions in the revision.
> >
> > I've also read the review by WPsh. I agree with WPsh that this paper needs some revision to improve its presentation and highlight its major contributions. Please do so accordingly. I'm still positive about this paper overall and my score remains unchanged.

---

### Official Review · Reviewer_D6xc · 2021-07-19

**Rating:** 7
**Confidence:** 4

**Summary:**

The paper presents a novel method for few-shot text classification tasks. The method is built on top of the CNAP  The key contribution is to effectively encode the gradients of networks as the task representation so that the meta-learning stage can use such task representations to improve the few-shot generalization ability. The two key technical contributions are a task embedding network (an RNN-based layer-by-layer gradient encoder via Fisher Information Matrix) and an adaptation network for auto-regressively adapting parameters from the bottom to the top layer.  The implementation is based on BERT-like LMs with Adapter bottleneck modules, which are parameter efficient to compute.

The authors compared the proposed method with a few meta-learning methods on a diverse set of tasks, and show that their method outperforms other methods for most of the time. The empirical analysis and ablations also provide insights for understanding the proposed method.




**Main Review:**


Overall, the work is a nice contribution towards using the gradient information to encode a task instead of only using the input/output text. The gradient information implies more about the network behavior which is closer to the features of a task. The paper is mostly well written with nice figures and tables for delivering important information. I do not have major concerns about the paper.  There are a few minor notes and suggestions below:

- I think in the problem definition, Line 98, it would be better to explicitly define the learning stages and a shared text classification model. The meta-learning stage has the access to the source tasks, and the few-shot generalization stage does not. And there is a shared model after the meta-learning stage.

- In the preamble of Section 4, it would be better to illustrate more the motivation of the design choice and define the "adaptation parameters" clearly. It is vague in the current presentation and makes people wonder about the underlying motivations of the design.

- It would be better to visualize the task representation in some way and show the task-to-task distances so that we can inspect if the closer tasks can locate near each other.

- Line 202, BERT-base only allows 128 tokens? I don't think so. It should be 512 if I understand correctly. Please double-check it and correct me if I was wrong about this.

- A few typos: Line 19: "a intermediate"; Line 201 BERT-BERT.

**Time Spent Reviewing:**

4

---

> ### Author Response · Authors · 2021-08-10
> **Thank you for your comments/suggestions. We will carefully address them in the revised version.**
>
> We thank the reviewer for the useful comments and suggestions.
>
> > Explicitly define the learning stages and a shared text classification model
>
> Thank you for the suggestion. In the problem definition, we will clearly define the learning stages and describe which parts of the model are shared for all tasks and which parts are task specific, and at which stages they are trained.
>
> > Describe motivation more clearly of design choices in the preamble of Section 4
>
> Thank you for the suggestion. We will add the following paragraph in the beginning of Section 4 to provide a high-level motivation, and also describe the motivation clearly when introducing every component:
>
> “To handle diverse tasks with different structures a single model may not be expressive enough. Thus, for every task, we utilize a task embedding network to capture the task nature and adapt the model conditioned on the current task. Task-specific adaptation is done by generating shifting and scaling parameters, named adaptation parameters, that are applied on the hidden representations inside the model.”
>
> > Visualize the task representation
>
> This is a great suggestion. We built visualizations showing that the learned task embeddings, mapped into a 2D space, form good task clusters, especially using the task embeddings learned on higher layers. We will include these figures in our paper.
>
> > Maximal sequence length allowed by BERT-base
>
> You are right. The BERT-base model actually allows sequence lengths up to 512. We will correct this in our paper. Since our major baseline Leopard restricts the maximal length to 128, we will keep this restriction for fair comparison with their results.
>
> Thanks for pointing out the typos. We have carefully revised the whole paper to correct all typos.

---

> > ### Comment · Reviewer_D6xc · 2021-08-31
> > **Comment about the rebuttal**
> >
> > Thank you for the response to my review. I have read it and other reviews and I'd like to keep my score based on the response. Please include the clarifications and improve the presentation in the next version of the paper. Thanks!

---

### Decision · Program_Chairs · 2021-09-27

**Decision:**

Accept (Poster)

**Comment:**

The reviewers all agreed that that the method of encoding gradients to obtain a task embedding and using FILM to adapt the text-classification network to that task. Many of the initial concerns raised primarily by reviewer WPsh were addressed in the rebuttal through additional experiments, leading the reviewer to raise their score by 1. In general, the design choices and claims are well supported by ablation studies and comparisons against various baselines on various datasets. The writing is also structured and of decent quality. That said, the authors need to address the new "puzzling" issues raised by reviewer WPsh's updated review on the fine-tuning and comparison to other methods, especially as they cast doubt on the validity of the reported numbers.